# *Salmonella* manipulates the host to drive pathogenicity via induction of interleukin 1β

**Mor Zigdon**[1], **Jasmin Sawaed**[1], **Lilach Zelik**[1], **Dana Binyamin**[1], **Shira Ben-Simon**[1], **Nofar Asulin**[1], **Rachel Levin**[1], **Sonia Modilevsky**[1], **Maria Naama**[1], **Shahar Telpaz**[1], **Elad Rubin**[1], **Aya Awad**[1], **Wisal Sawaed**[1], **Sarina Harshuk-Shabso**[1], **Meital Nuriel-Ohayon**[1], **Mathumathi Krishnamohan**[2], **Michal Werbner**[1], **Omry Koren**[1], **Sebastian E. Winter**[3], **Ron N. Apte**[2†], **Elena Voronov**[2], **Shai Bel**[1]*

1 Azrieli Faculty of Medicine, Bar-Ilan University, Safed, Israel, 2 The Shraga Segal Department of Microbiology, Immunology and Genetics, Faculty of Health Sciences, Ben Gurion University of the Negev, Beer Sheva, Israel, 3 Department of Internal Medicine, Division of Infectious Diseases, UC Davis Health, Davis, California, United States of America

† Deceased.
* shai.bel@biu.ac.il

**Data Availability Statement:** All RNA and microbial 16S sequencing data is available at NCBI GEO GSE252071. All data underlying the main and

## Abstract

Acute gastrointestinal infection with intracellular pathogens like *Salmonella* Typhimurium triggers the release of the proinflammatory cytokine interleukin 1β (IL-1β). However, the role of IL-1β in intestinal defense against *Salmonella* remains unclear. Here, we show that IL-1β production is detrimental during *Salmonella* infection. Mice lacking IL-1β (*IL-1β⁻/⁻*) failed to recruit neutrophils to the gut during infection, which reduced tissue damage and prevented depletion of short-chain fatty acid (SCFA)-producing commensals. Changes in epithelial cell metabolism that typically support pathogen expansion, such as switching energy production from fatty acid oxidation to fermentation, were absent in infected *IL-1β⁻/⁻* mice which inhibited *Salmonella* expansion. Additionally, we found that IL-1β induces expression of complement anaphylatoxins and suppresses the complement-inactivator carboxypeptidase N (CPN1). Disrupting this process via IL-1β loss prevented mortality in *Salmonella*-infected *IL-1β⁻/⁻* mice. Finally, we found that *IL-1β* expression correlates with expression of the complement receptor in patients suffering from sepsis, but not uninfected patients and healthy individuals. Thus, *Salmonella* exploits IL-1β signaling to outcompete commensal microbes and establish gut colonization. Moreover, our findings identify the intersection of IL-1β signaling and the complement system as key host factors involved in controlling mortality during invasive Salmonellosis.

## Introduction

The cytokine interleukin 1β (IL-1β) is a proinflammatory alarmin that is released mainly from activated myeloid cells during acute and chronic inflammation. To facilitate its rapid secretion, IL-1β is stored in the cell as a propeptide which is cleaved to its mature form by Caspase 1 following activation of the inflammasome by invasive bacteria. While inflammasome-mediated

supplementary figures can be found at supplementary S1 Data.

**Funding:** This work was supported by the Azrieli Foundation Early Career Faculty Fellowship (to SB), the Israeli Science Foundation (ISF) (925/19 and 1851/19 to SB), the European Research Council (ERC) Starting Grant (GCMech 101039927 to SB) and the U.S-Israel Binational Science Foundation (2021025 to SEW and SB). The funders had no role in study design, data collection and analysis, decision to publish, or preparation of the manuscript.

**Competing interests:** The authors have declared that no competing interests exist.

**Abbreviations:** ANCOM, analysis of composition of microbiome; CBC, complete blood count; CPN1, carboxypeptidase N; d.p.i., days post-infection; ICU, intensive care unit; LPS, lipopolysaccharide; M.L.N., mesenteric lymph node; MOI, multiplicity of infection; MPO, myeloperoxidase; NE, neutrophil elastase; PCoA, principal coordinate analysis; SCFA, short-chain fatty acid; TLR, toll-like receptor; WT, wild-type.

secretion of IL-1β is well characterized, IL-1β is also known to be secreted via inflammasome-independent mechanisms [1]. Once secreted, IL-1β affects multiple cell types via binding to the IL-1 receptor [2]. Notably, IL-1β secretion affects endothelial cell permeability to allow massive infiltration of neutrophils from the periphery into the infected tissue [3]. While these activated neutrophils contribute to bacterial killing, they also have a damaging effect on host tissues.

The foodborne pathogen *Salmonella* enterica serovar Typhimurium (hereafter *Salmonella*) is a common cause of acute gastrointestinal inflammation, caused by consumption of contaminated food. While this infection is usually self-limiting in healthy humans, it can lead to life-threatening bacteremia in immune-compromised individuals. In mice carrying a mutated *Nramp1* gene, such as all mice on a C57BL/6 background, oral infection leads to acute colitis and systemic dissemination of *Salmonella*, ultimately resulting in death within several days [4]. The *Nramp1* gene encodes an ion channel expressed in macrophages and it is thought that loss of this gene renders phagocytic cells unable to control growth of intracellular bacteria [5]. Yet, the mechanism leading to mortality caused by *Salmonella* infection in mice is not clear.

When *Salmonella* is ingested via contaminated food it reaches the host intestine, where it is confronted by the gut microbiota. These microbes are well adapted to the unique environment in the gut and are thus well suited to outcompete *Salmonella* for the limited resources available in the gut [6]. Commensal microbes also maintain a symbiotic relationship with the host, where dietary fibers from the host diet are fermented to produce short-chain fatty acids (SCFAs) by the microbiota, mainly by bacteria from the Clostridia class [7]. These SCFA are used as an energy source by host colonocytes via fatty-acid oxidation, a process which utilizes oxygen in the tissue, thus leaving the gut lumen hypoxic [7]. To gain a foothold in the gut, *Salmonella* influences the host to drive remodeling of the gut niche through inflammation [8], yet exactly how this process occurs is not clear.

Studies in mice have revealed that *Salmonella* can exploit the host's innate immune responses to allow its own expansion. Mice lacking toll-like receptors (TLRs) 2, 4, and 9 are more resistant to *Salmonella* infection than mice lacking either TLRs 2 and 4 or 4 and 9 alone, indicating that bacterial sensing by the host can be detrimental [9]. In another example, mice lacking the cytokine IL-22 were found to limit *Salmonella* luminal growth better than wild-type mice (WT), as secretion of lipocalin-2 and calprotectin under the control of IL-22 suppressed commensal microbes which directly compete with *Salmonella* [10]. Thus, *Salmonella* has evolved to exploit the host's defense mechanisms for its own advantage [11]. Here, we set out to determine whether host production of IL-1β is protective during *Salmonella* infection and discovered that the pathogen exploits this cytokine to drive pathogenicity.

## Results

### Loss of IL-1β in mice reduces *Salmonella* burden and prevents mortality during oral infection

Previous studies have shown that IL-1β has a protective role during chemical-induced colitis, by driving epithelial repair following injury [12]. To determine whether IL-1β is required for host defense during infection-induced colitis, we challenged streptomycin-treated WT and *IL-1β* $^{-/-}$ mice (on a C57BL/6 background; *Nramp1*$^s$ S1 Fig) with a single oral dose of *Salmonella*. Pretreatment with streptomycin is crucial to drive robust gastrointestinal inflammation [13]. We found that *Salmonella* levels were reduced 100-fold in the cecal lumen of *IL-1β* $^{-/-}$ mice 4 days post-infection (d.p.i.) compared to WT mice (Fig 1A). *IL-1β* $^{-/-}$ mice also harbored 10-fold less *Salmonella* in gut-draining mesenteric lymph nodes (M.L.N.), spleen, and liver compared to WT mice (Fig 1B–1D). This reduction in bacterial loads in *IL-1β* $^{-/-}$ mice was

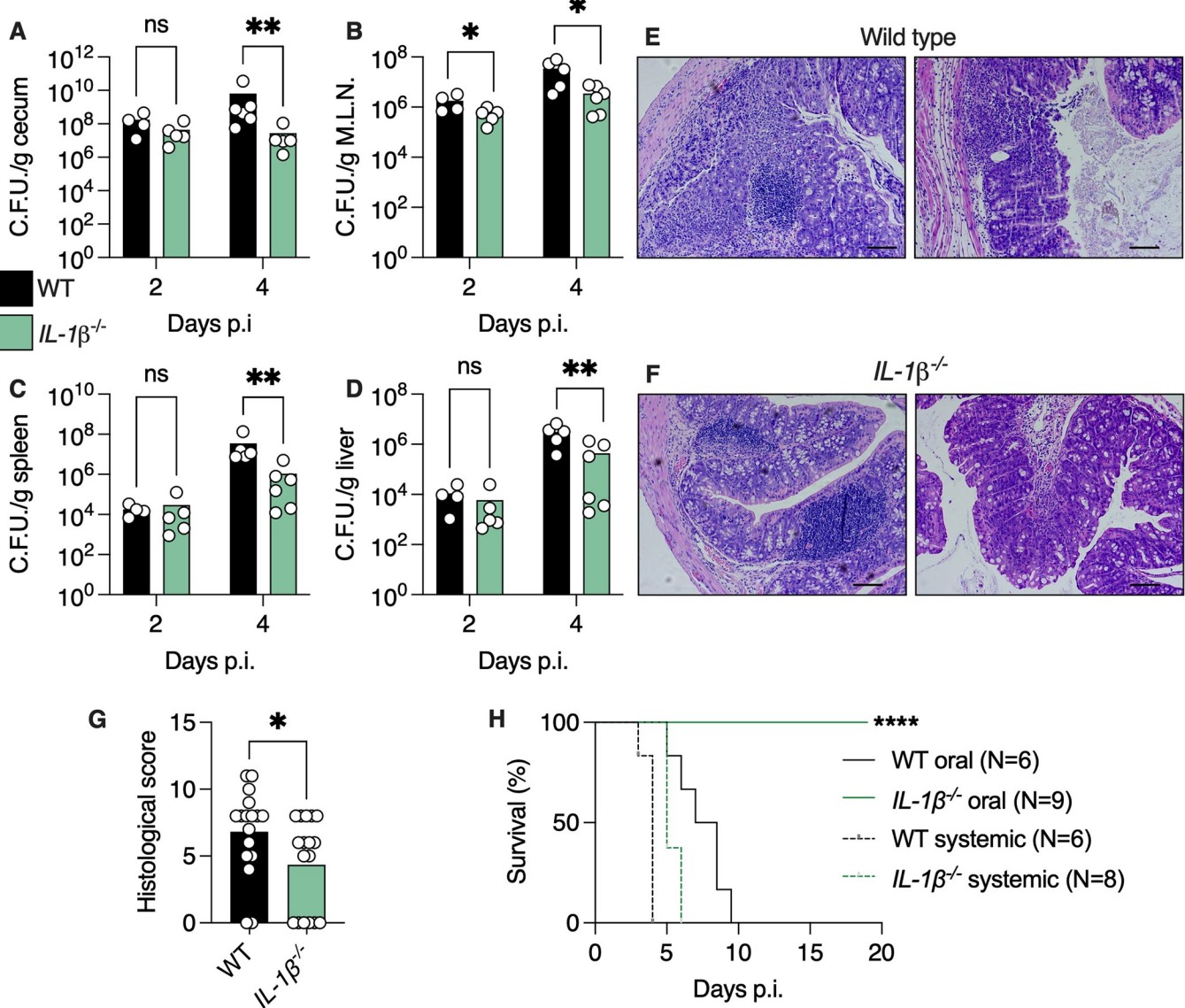

**Fig 1. Loss of IL-1β dampens *Salmonella* growth in vivo and prevents mortality from oral infection.** (**A–D**) *Salmonella* C.F.U. in cecal content (**A**), M.L.N. (**B**), spleen (**C**), and liver (**D**) of mice infected with $10^7$ C.F.U. *Salmonella* enterica serovar typhimurium (SL1344) 24 h after pretreatment with 20 mg streptomycin. (**E, F**) Representative histological images of colonic tissue of WT (**E**) and *IL-1β*$^{-/-}$ (**F**) mice 2 d.p.i. (**G**) Histological damage analysis of mice as in **E** and **F**. (**H**) Survival percentage of mice infected orally or intravenously (systemic infection) with *Salmonella*. (**A–D** and **G**) Each dot represents a mouse. *$P < 0.05$; **$P < 0.01$; ****$P < 0.0001$. (**A–D** and **G**) Mann–Whitney test. (**H**) Mantel–Cox test. (**E** and **F**) Scale bar, 100 μm. These data are representative of at least 4 independent experiments. Numerical values are in S1 Data. C.F.U., colony-forming units; d.p.i., days post-infection; M.L.N., mesenteric lymph node; WT, wild-type.

accompanied by a marked reduction in histological colonic tissue damage (Fig 1E–1G). The composition of the gut microbiota before infection can affect the ability of *Salmonella* to thrive in the intestine [14]. To determine whether preinfection differences in microbiota composition between WT and *IL-1β*$^{-/-}$ mice affect *Salmonella* colonization, we performed co-housing experiments (to normalize the microbiota composition [15]) and found that co-housing did not lead to different results than separately housed mice (S2A–S2D Fig). Thus, loss of IL-1β limits *Salmonella* expansion in the gut lumen and systemic sites.

We then tested whether loss of *IL-1β* affects infection-induced mortality. Remarkably, we found that *IL-1β* $^{-/-}$ mice were completely resistant to infection-induced mortality, while WT mice all succumbed to the infection (Fig 1H). This resistance of *IL-1β* $^{-/-}$ mice was also present in co-housed WT and *IL-1β* $^{-/-}$ mice (S2E Fig), indicating that microbiota differences before infection did not affect mortality in these experiments. To determine whether this resistance to infection is dependent on mode of infection, we infected mice intravenously with *Salmonella*. We found that *IL-1β* $^{-/-}$ mice infected intravenously with a low dose of *Salmonella* die from the infection in a similar manner as WT mice (Fig 1H). These results indicate that IL-1β plays a role in mortality during oral *Salmonella* infection.

## IL-1β $^{-/-}$ mice fail to recruit neutrophil to the gut during *Salmonella* infection

We next wanted to determine how IL-1β production drives *Salmonella* expansion in the gut lumen. Mice carrying the mutant *Nramp1* allele, such as C57BL/6 mice, are thought to fail in controlling *Salmonella* expansion because of defective macrophage function [4]. Thus, we hypothesized that IL-1β impairs intracellular killing of *Salmonella* by macrophages. To test this, we extracted peritoneal macrophages from WT and *IL-1β* $^{-/-}$ mice and tested their ability to kill intracellular *Salmonella* in vitro. Contrary to our hypothesis, we found that macrophages from *IL-1β* $^{-/-}$ mice were less efficient at killing intracellular *Salmonella* (S3A Fig). This result indicates that loss of IL-1β impairs *Salmonella* growth in a manner which is not dependent on intracellular killing by macrophages.

To understand how IL-1β deficiency affects the colonic tissue during *Salmonella* infection, we performed bulk RNA sequencing followed by pathway analysis of differentially expressed genes. We found that genes which were down-regulated in *IL-1β* $^{-/-}$ mice are involved in neutrophil function and recruitment to the colon (Fig 2A). Indeed, key chemokines that attract neutrophil and monocytes to the colon, along with neutrophil-specific transcripts such as lipocalin-2 and calprotectin, were expressed at lower levels in infected *IL-1β* $^{-/-}$ mice (Fig 2B). These differences were absent in uninfected WT and *IL-1β* $^{-/-}$ mice (S4 Fig). We validated these findings by counting myeloperoxidase (MPO)- and neutrophil elastase (NE)-positive cells in colonic tissue sections from infected mice. Accordingly, *IL-1β* $^{-/-}$ mice had on average 4-fold fewer neutrophils in their colons (Fig 2C–2E). We then tested whether loss of IL-1β leads to reduced neutrophils levels during *Salmonella* infection because of neutrophil development defects, or failure to recruit neutrophils to the colonic tissue. By analyzing circulating blood from naïve and infected mice, we found that loss of IL-1β does not affect circulating neutrophil levels during steady state, but rather prevents neutrophil recruitment to circulation during infection (Fig 2F and 2G). Thus, the resistance of *IL-1β* $^{-/-}$ mice to *Salmonella* is not due to an elevated antimicrobial response as they fail to draw neutrophils to the site of infection.

## IL-1β leads to collapse of gut short-chain fatty acid-producing Clostridia during *Salmonella* infection

Commensal gut microbes provide colonization resistance by directly inhibiting pathogen growth and by affecting the host [16]. We next tested whether changes in the gut microbiota of *IL-1β* $^{-/-}$ mice can explain their resistance to *Salmonella* infection. We reasoned that lack of neutrophil recruitment to the gut would dampen the harmful damage of infection-induced inflammation on the microbiota [16]. 16S rRNA microbiota analysis of feces from infected WT and *IL-1β* $^{-/-}$ mice revealed that the latter contained a more diverse microbiota after *Salmonella* infection (Fig 3A). Jaccard analysis confirmed that the gut microbial composition of infected *IL-1β* $^{-/-}$ mice was distinct from that of WT mice (Fig 3B). Analysis of composition of

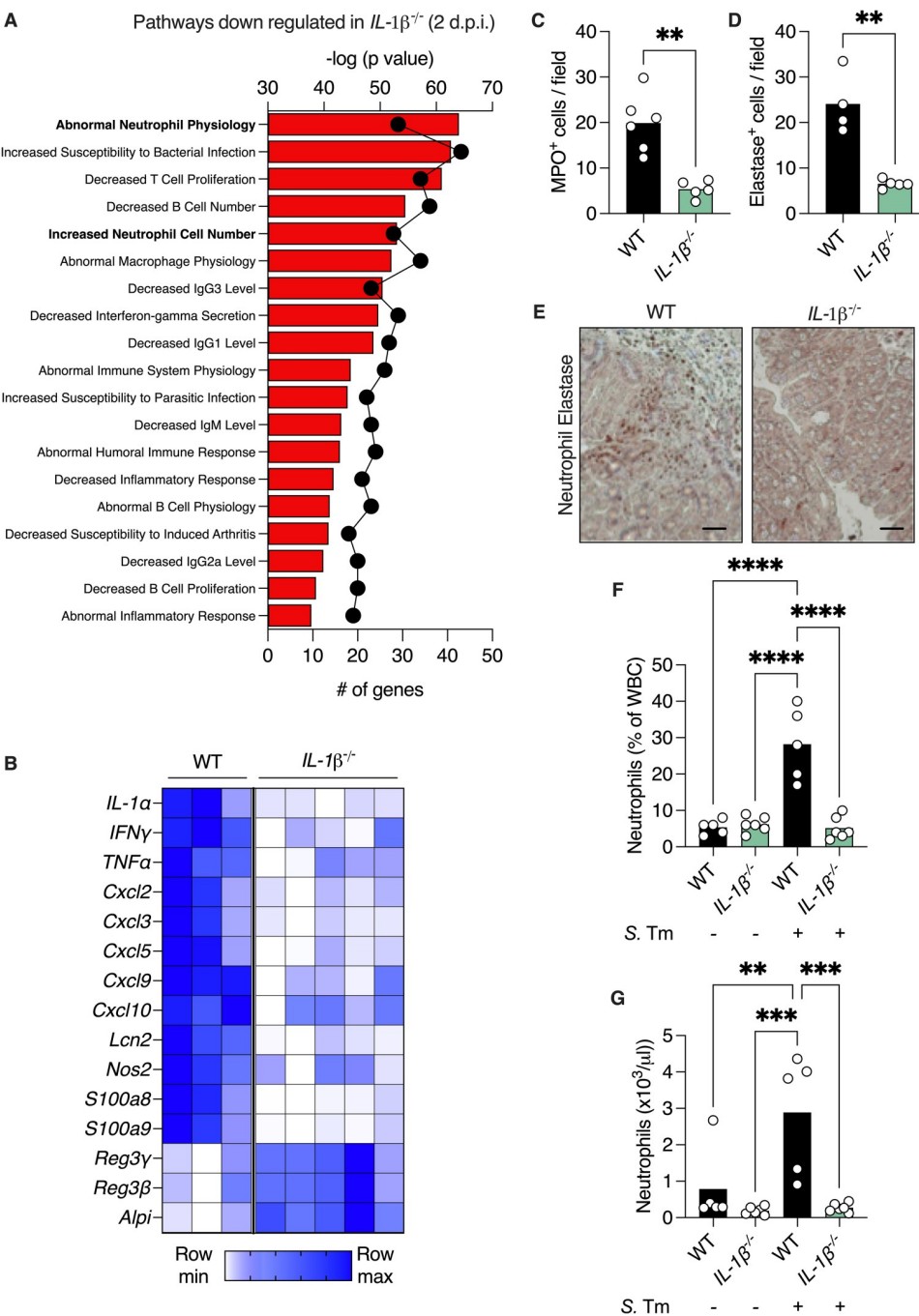

**Fig 2. Loss of IL-1β impairs neutrophil recruitment to the gut during *Salmonella* infection.** (**A**) Pathway analysis of transcripts that are down-regulated in colonic tissue of *Salmonella*-infected *IL-1β* <sup>-/-</sup> according to GO biological function. Bars represent -log (*P* value) and dots represent number of genes in pathway. (**B**) Heatmap depicting differentially expressed innate immune genes with a *P* < 0.05. Each column represents a mouse and each row a gene. (**C, D**) Numbers of NE-positive (**C**) and myeloperoxidase-positive (**D**) cells in colonic section of *Salmonella*-infected mice 4 d.p.i. Each dot represents a mouse. (**E**) Representative immunohistochemistry images of colonic section from *Salmonella*-infected mice stained with anti-neutrophil elastase antibody. Scale bar, 50 μm. (**F**) % of neutrophils out of total circulating WBC and (**G**) concentration of neutrophils in serum of WT and *IL-1β* <sup>-/-</sup> mice infected as indicated 4 d.p.i. *$P$ < 0.05; **$P$ < 0.01; ***$P$ < 0.001; ****$P$ < 0.0001. (**C** and **D**) Student's *t* test. (**F** and **G**) One-way ANOVA. d. p.i., days post-infection; *S.* Tm, *Salmonella* typhimurium. These data are representative of 2 independent experiments. Numerical values are in S1 Data. The underlying data for this figure can be found at GSE252071. NE, neutrophil elastase; WT, wild-type.

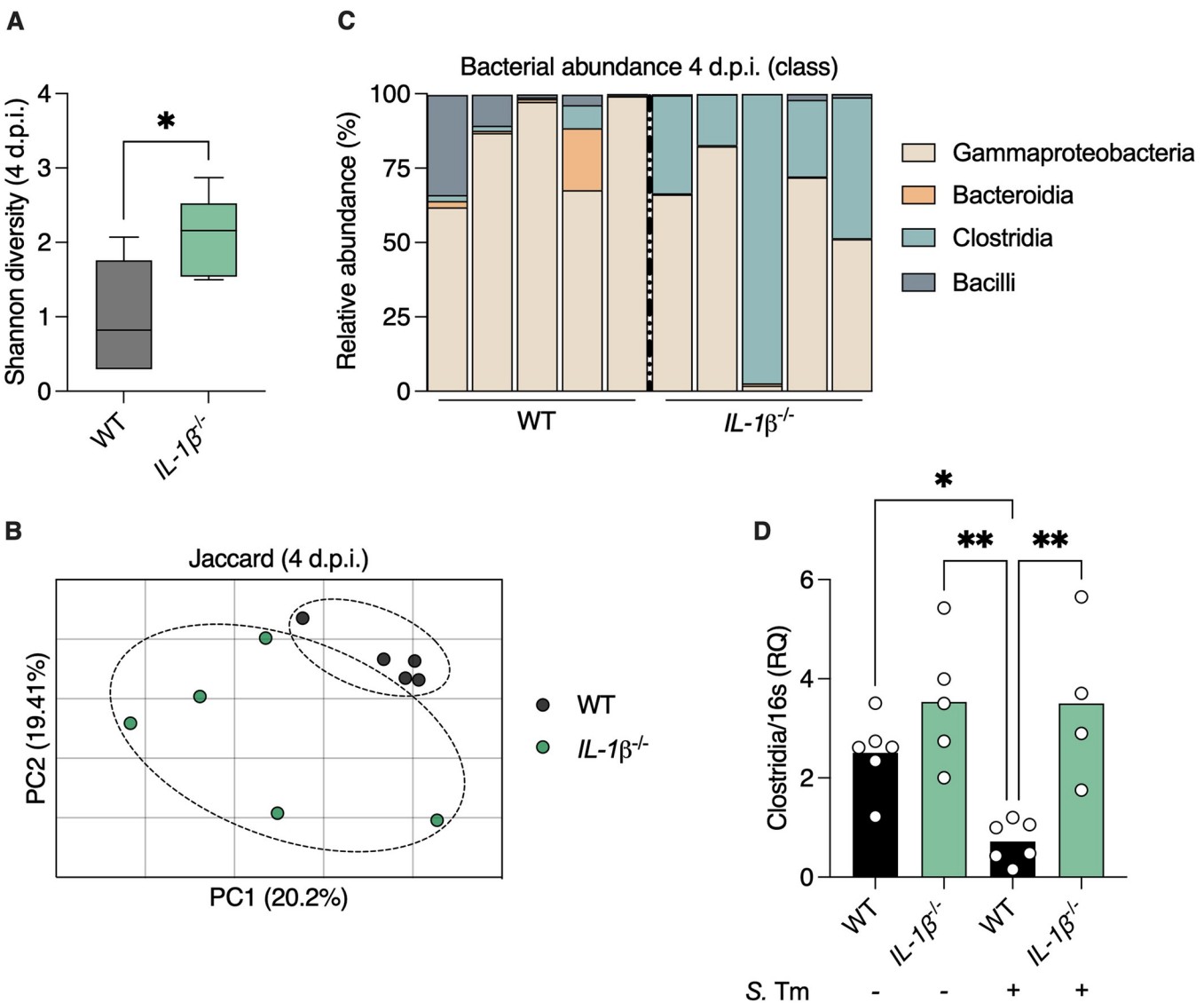

**Fig 3. *Salmonella* infection does not deplete SCFA-producing Clostridia from the gut of *IL-1β* <sup>-/-</sup> mice.** 16 S rRNA sequencing was performed to characterize gut microbiota composition. (**A**) Shannon index representing microbial diversity in gut microbiota of *Salmonella*-infected mice. (**B**) PcoA of fecal microbiota based on Jaccard similarity coefficient. Each dot represents a mouse. (**C**) Relative taxonomic composition at the class levels. Each column represents a mouse. (**D**) qPCR analysis of levels of the class Clostridia in naïve or infected mice as indicated. Each dot represents a mouse. $*P < 0.05$; $**P < 0.01$; (**A**) Student's *t* test, (**D**) one-way ANOVA. d.p.i., days post-infection; *S*. Tm, *Salmonella* typhimurium. These data are representative of at least 2 independent experiments. Numerical values are in S1 Data. The underlying data for this figure can be found at GSE252071. PCoA, principal coordinate analysis; SCFA, short-chain fatty acid.

microbiomes (ANCOM) showed that bacteria from the Clostridia class were enriched in the gut of infected *IL-1β* <sup>-/-</sup> mice while these microbes were virtually absent from the gut of infected WT mice (Figs 3C and S5A). Previous reports have shown that *Salmonella* infection leads to depletion of SCFA-producing Clostridia microbes [7]. We found that in naïve WT and *IL-1β* <sup>-/-</sup> mice, the levels of Clostridia bacteria were similar. However, *Salmonella* infection dramatically reduced the levels of Clostridia bacteria in infected WT mice while not affecting these microbes in *IL-1β* <sup>-/-</sup> mice (Fig 3D). This post-infection reduction in levels of Clostridia were not dependent on preinfection conditions as co-housing and streptomycin treatment of WT and *IL-1β* <sup>-/-</sup> mice did not result in differences in Clostridia levels (S5B Fig). Thus,

*Salmonella* infection does not lead to depletion of SCFA-producing Clostridia bacteria in *IL-1β* $^{-/-}$ mice.

## Preservation of SCFA-producing bacteria in IL-1β $^{-/-}$ mice inhibits *Salmonella* expansion by maintaining beta oxidation in colonocytes and hypoxic condition in the gut

Next, we wanted to determine how this different microbiota in infected *IL-1β* $^{-/-}$ mice affects the colonic tissue. We found that genes which were up-regulated in the colon of infected *IL-1β* $^{-/-}$ mice were enriched in metabolism pathways (Fig 4A). Specifically, we found that genes encoding proteins that participate in energy production via fatty acid beta oxidation were up-regulated in infected *IL-1β* $^{-/-}$ mice as compared to infected WT mice (Fig 4B). These differences were not present in uninfected WT and *IL-1β* $^{-/-}$ mice (S6A and S6B Fig).

SCFAs produced by commensal Clostridia bacteria serve as the preferred energy source for colonocytes. These epithelial cells produce ATP via beta oxidation of SCFA, which renders the colonic lumen hypoxic as this process utilizes oxygen. *Salmonella* infection has been shown to deplete SCFA-producing bacteria from the colon, thus forcing colonocytes to produce energy via glycolysis [7]. As this process does not require oxygen, *Salmonella* infection effectively leads to elevated oxygen levels in the colonic lumen. Oxygen in the lumen is toxic to the obligatory anaerobic commensal microbiota, which is beneficial for *Salmonella* as it disrupts the colonization resistance provided by these commensals [16]. Indeed, we found that all the genes in the glycolysis pathway were elevated in infected WT compared to *IL-1β* $^{-/-}$ mice (Fig 4C). These differences were not present in uninfected WT and *IL-1β* $^{-/-}$ mice (S6C–S6E Fig). Accordingly, we found that the colonocytes in *IL-1β* $^{-/-}$ mice remained hypoxic during *Salmonella* infection while colons of WT mice became rich in oxygen (Fig 4D and 4E).

These observations led us to hypothesize that failure to recruit neutrophils in *IL-1β* $^{-/-}$ mice preserves hypoxic conditions in the colon by sparing SCFA-producing Clostridia bacteria. This in turn can inhibit *Salmonella* growth by providing direct competition and by reducing oxygen levels that are important for *Salmonella* growth [16]. To test our hypothesis, we depleted Clostridia bacteria from the colon of WT and *IL-1β* $^{-/-}$ mice by vancomycin treatment (in addition to the standard streptomycin treatment) and then infected these mice with *Salmonella*. We validated that vancomycin treatment indeed depleted the major SCFA-producing members of the Clostridia class [17] via 16S sequencing (S7 Fig). Indeed, we found that *Salmonella* growth in the intestine of *IL-1β* $^{-/-}$ mice was indistinguishable from WT mice after vancomycin treatment (Fig 4F). Thus, preservation of SCFA-producing bacteria in *IL-1β* $^{-/-}$ mice maintains energy production via beta oxidation and hypoxic condition in the gut which in turn inhibit *Salmonella* growth.

## IL-1β drives mortality in *Salmonella*-infected mice by suppressing the anaphylatoxin-inactivator carboxypeptidase N

Next, we wanted to determine whether preserving SCFA-producing Clostridia is the mechanism that prevents mortality in *Salmonella*-infected *IL-1β* $^{-/-}$ mice. However, we found that vancomycin-treated *IL-1β* $^{-/-}$ mice still survived after oral *Salmonella* infection (S8A Fig). We then hypothesized that the reason for resistance to *Salmonella* infection in *IL-1β* $^{-/-}$ mice is that they can clear the infection. Yet, we found that even 17 d.p.i. these *IL-1β* $^{-/-}$ mice were still colonized by *Salmonella* (S8B Fig), and that the pathogen was still fully virulent as transmission to WT hosts was still lethal (S8C Fig).

To understand why *IL-1β* $^{-/-}$ mice do not die from *Salmonella* infection, we performed RNA sequencing on mice 6 d.p.i., as this is the time point when WT mice are moribund. We

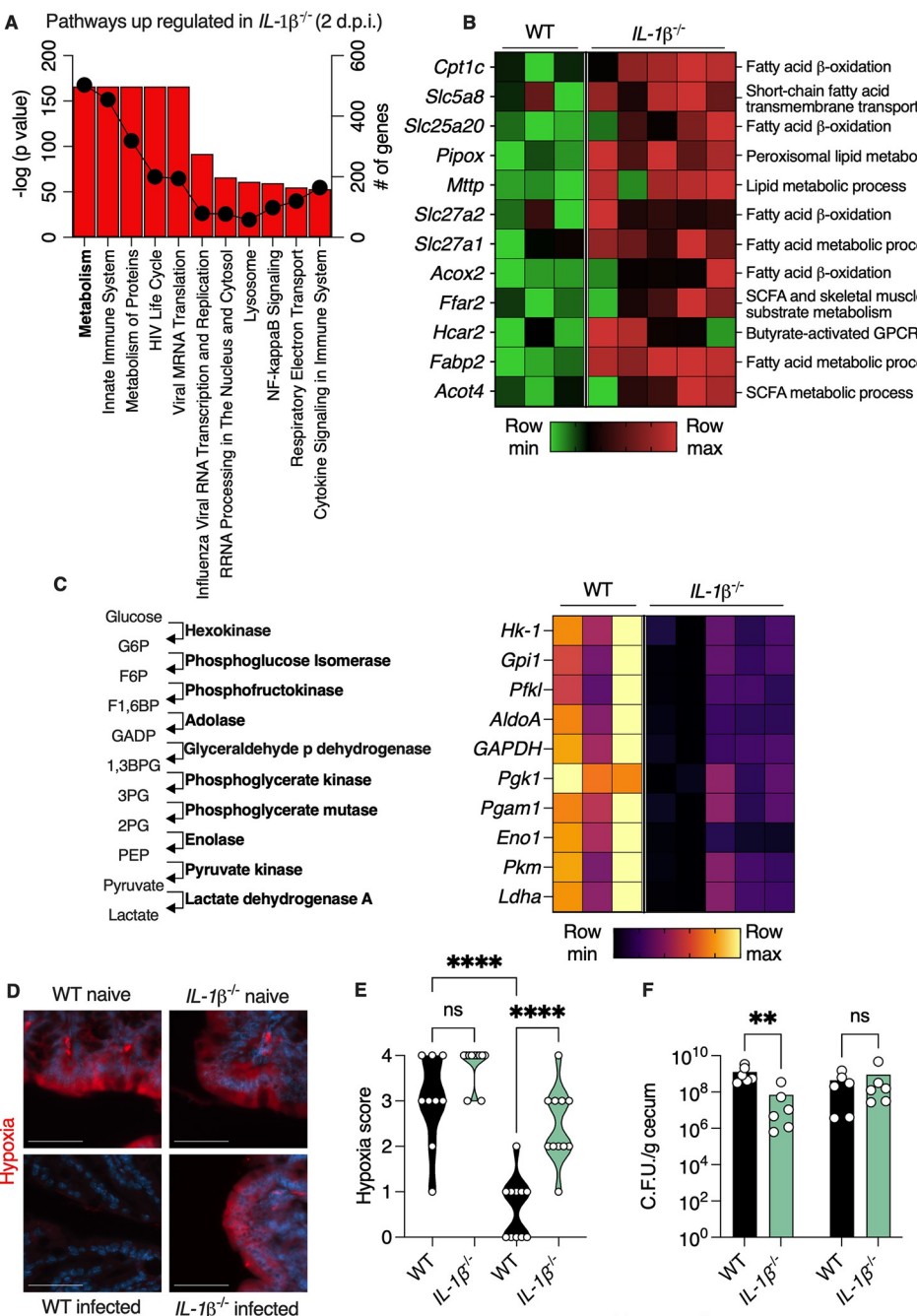

**Fig 4. Preservation of fatty acid beta-oxidation in *IL-1β* -/- mice inhibits *Salmonella* growth in vivo. (A)** Pathway analysis of transcripts that are up-regulated in colonic tissue of *Salmonella*-infected *IL-1β* -/- according to GO biological function. Bars represent -log (*P* value) and dots represent number of genes in pathway. **(B)** Heatmap depicting differentially expressed genes involved in fatty acid oxidation with a *P* < 0.05. Each column represents a mouse and each row a gene. **(C)** Heatmap depicting differentially expressed genes in the glycolysis pathway with a *P* < 0.05. The enzymatic activity of each gene in the glycolysis pathway is presented on the left. Each column represents a mouse and each row a gene. **(D)** Immunofluorescence microscopy of colonic section from mice. Red staining shows pimonidazole that indicates hypoxia levels. Nuclei were stained with DAPI. Scale bar, 50 μm. **(E)** Quantification of red signal in **(D)**. Each dot represents a mouse. **(F)** *Salmonella* C.F.U. in cecal contents of infected mice 4 d.p.i. treated as indicated. Each dot represents a mouse. **P < 0.01, ****P < 0.0001; **(E)** one-way ANOVA; **(F)** Mann–Whitney test. d. p.i., days post-infection; *S*. Tm, *Salmonella* typhimurium. These data are representative of 2 independent experiments. Numerical values are in S1 Data. The underlying data for this figure can be found at GSE252071.

reasoned that genes that are differently expressed in WT mice 6 d.p.i. compared to WT mice 2 d.p.i., and to *IL-1β* [-/-] mice 6 d.p.i., are related to the process that leads to mortality. We excluded genes that are differently expressed in *IL-1β* [-/-] mice 6 d.p.i. versus 2 d.p.i. as these mice do not die from the infection, thus these genes are not related with mortality (Fig 5A). Pathway analysis revealed that WT mice express genes that are related to vasculature morphology 6 d.p.i. (Fig 5A). We found this interesting, as septic shock which occurs during systemic infection by pathogens is characterized by vasculature permeability, dilation, and a severe hypotensive response [18,19]. These harmful changes to the vasculature are mediates by complement proteins, also known as anaphylatoxins, which can affect vasculature permeability at sub-nanomole concentrations [18]. We found that transcription levels of the complement *C3* gene and the complement receptor *C3ar1* gene were highly induced in WT mice at 6 d.p.i. (Fig 5B and 5C). While these genes were also induced in *IL-1β* [-/-] mice, their levels at 2 and 6 d.p.i. were still significantly lower than in WT mice (Fig 5B and 5C). Thus, IL-1β drives expression of anaphylatoxin proteins and receptors during *Salmonella* infection.

Under normal conditions, the damaging effects of anaphylatoxins are balanced by carboxypeptidase N (CPN1), which inactivates anaphylatoxins by peptide cleavage within seconds at physiological concentrations [20,21]. We found that *IL-1β* [-/-] mice express 4-fold higher transcription levels of *Cpn1* after *Salmonella* infection, compared with WT mice (Fig 5D). Indeed, a previous report has shown that IL-1 receptor signaling inhibits *Cpn1* expression [22]. These results imply that low expression and inactivation of the complement system in *IL-1β* [-/-] mice preserves their viability during infection. To test this, we treated mice with a CPN1 inhibitor. We found that inhibiting CPN1 in uninfected mice does not affect their viability (S9A Fig). In infected WT mice, administration of CPN1 inhibitor did not affect survival (Fig 5E). However, infected *IL-1β* [-/-] mice treated with CPN1 inhibitor suffered from over 60% mortality, while vehicle-treated *IL-1β* [-/-] mice were completely resistant to infection-induced mortality (Fig 5E). We also verified that the inhibitor does not affect *Salmonella* growth (S9B Fig). Thus, loss of IL-1β protects mice from *Salmonella*-induced mortality by removing suppression from CPN1 expression.

## Expression of complement 3 receptor correlates with IL-1β expression in sepsis patients

Finally, we wanted to determine whether the link between the complement system, IL-1β and life-threatening infection, which we show here in mice, is also relevant for human disease. We first analyzed a publicly available whole-blood transcriptional dataset from healthy individuals that underwent experimental endotoxemia via intravenous injection of lipopolysaccharide (LPS) [23]. We found that LPS administration led to an increase in expression levels of *IL-1β*, *C3AR1*, and the IL-1β receptor *IL1R1* transcripts (Fig 6A–6C). Additionally, we found that expression of *IL-1β* positively correlated with expression of *C3AR1* in individuals treated with LPS, but not untreated controls (Fig 6D and 6E). Thus, endotoxemia leads to positive correlation between expression of *IL-1β* and complement receptor gene *C3AR1* in humans.

Next, we analyzed peripheral blood RNA sequencing data from a cohort of sepsis patients [24]. We found higher levels of *IL-1β*, *C3AR1*, and *IL1R1* transcripts in patients with sepsis (Fig 6F–6H) and a positive correlation between expression levels of *IL-1β* and *C3AR1* in sepsis patients but not healthy controls (Fig 6I and 6J). Finally, we analyzed whole-blood transcriptional data from a cohort that included patients admitted to the intensive care unit (ICU) with sepsis, patients admitted to the ICU with non-infectious conditions, and healthy controls [25]. We found positive correlation between expression of *IL-1β* and *C3AR1* only in patients suffering from sepsis (Fig 6K–6M). Together, these results indicate that expression of *IL-1β*, the IL-1β receptor, and the

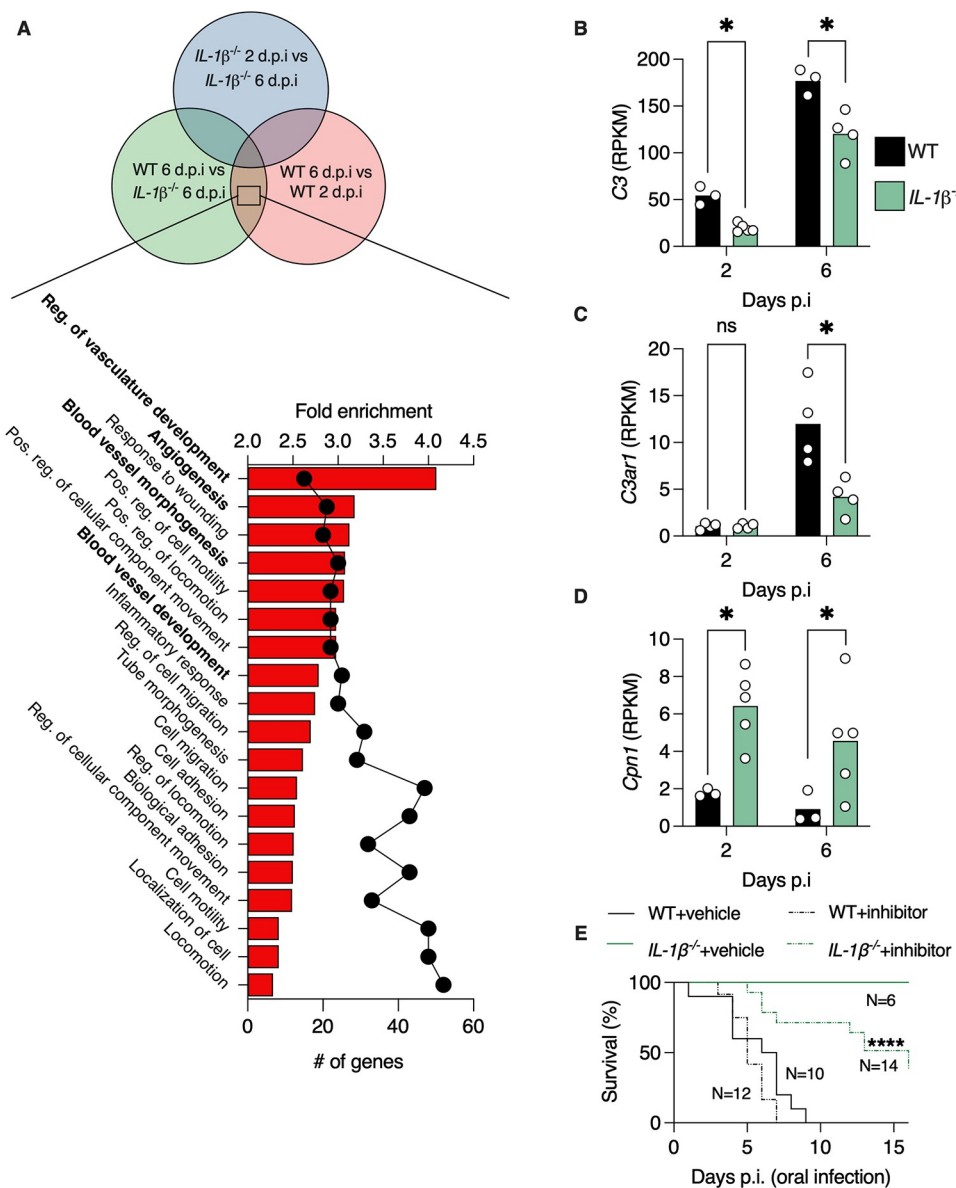

**Fig 5. IL-1β promotes expression of anaphylatoxin and inhibits expression of carboxypeptidase N that drives mortality in *Salmonella*-infected mice.** (**A**) Pathway analysis of transcripts that are differently expressed as indicated in Venn diagram in colonic tissue of *Salmonella*-infected mice according to GO biological function. Bars represent fold enrichment and dots represent number of genes in pathway. (**B–D**) Normalized reads of the indicated genes from colons of mice based on RNA sequencing. Each dot represents a mouse. (**E**) Survival of *Salmonella*-infected mice treated with vehicle or CPN1 inhibitor. \*$P < 0.05$, \*\*\*\*$P < 0.0001$. (**B–D**) Student's *t* test. (**E**) Mantel–Cox test. d.p.i., days post-infection; CPN, carboxypeptidase N. These data are representative of 2 independent experiments. Numerical values are in S1 Data. The underlying data for this figure can be found at GSE252071.

complement 3 receptor transcripts are induced in humans suffering from sepsis and that positive correlation between expression of *IL-β* and *C3AR1* occurs only in sepsis patients.

## Discussion

*Salmonella* has evolved to be a master manipulator of host response. By driving acute inflammation in the gut, *Salmonella* facilitates the killing of its microbial competitors via the host.

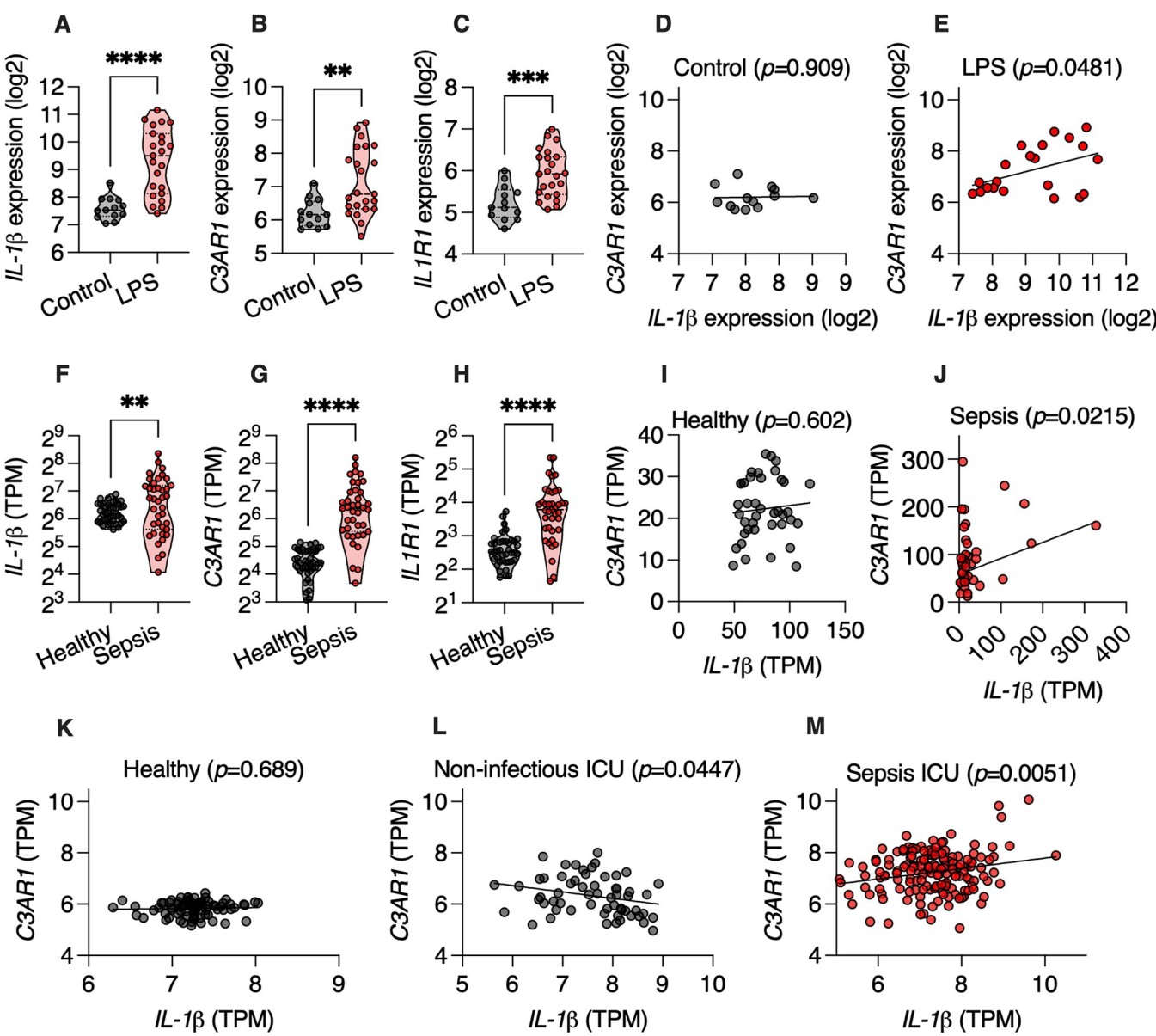

**Fig 6. Expression of *IL-1β* correlates with expression of the C3 receptor gene in sepsis patients.** (**A–E**) Analysis of whole-blood transcriptional microarray data from healthy humans treated intravenously with 2 ng/kg *Escherichia coli* lipopolysaccharide (LPS) (GSE134356). (**F–J**) Analysis of whole-blood RNA sequencing data from hospitalized sepsis patients and healthy controls (GSE154918). (**K–M**) Analysis of whole-blood transcriptional microarray data from patients admitted to the ICU with sepsis, without sepsis, or healthy controls (GSE134347). Each dot represents a patient. **$P < 0.01$; ***$P < 0.001$; ****$P < 0.0001$. (**A–C** and **F–H**) Student's *t* test. (**D**, **E**, and **I–M**) Simple linear regression. TPM, transcripts per million. Numerical values are in S1 Data. ICU, intensive care unit; LPS, lipopolysaccharide.

Indeed, recent research has shown that, under experimental settings, an immune-compromised host (through loss of TLR signaling, IL-22 production, or STAT2 signaling) is better equipped to contain *Salmonella* infection [9,10,26]. Here, we show that IL-1β is a central component of host manipulation by *Salmonella*. Using IL-1β signaling to facilitate its colonization is an effective mechanism deployed by *Salmonella*, as multiple microbial detection apparatuses converge on processing and release of IL-1β [1,27]. Indeed, secretion of IL-1β can be performed even in the absence of inflammasome activation [1] and without resulting in

pyroptosis [28]. This capability of different pathways to drive IL-1β secretion can also explain why mice deficient in individual components of the inflammasome have not shown the same resistance to *Salmonella* infection as we show here in *IL-1β* $^{-/-}$ mice [29]. Along these lines, a recent report has shown that only combined ablation of all the pathways which drive NLRC4 inflammasome-dependent cell death (deletion of *Asc*, *Caspase 1* and *Caspase 11*, simultaneously) can protect mice from fatal *Salmonella* infection [30]. These studies, along with the data presented here, indicate that deletion or inhibition of specific inflammasomes or Caspases does not result in complete loss of IL-1β production. Thus, it is possible that *Salmonella* has evolved to depend on IL-1β for host colonization, as infection consistently results in IL-1β secretion, even if not all pathogen-detection mechanisms are triggered.

Multiple studies have focused on the role of inflammasomes in host response to *Salmonella* infection [29]. Most of these studies have focused on the ability of inflammasome activation to limit bacterial growth in inner tissues, such as liver and spleen. Thus, we chose to focus our study on the role that IL-1β plays in remodeling of the intestinal niche, rather than the mode of its production and release. Our results highlight the exploitation of IL-1β by *Salmonella* to establish a foothold in the luminal niche. We show that IL-1β production in response to infection activates a cascade of events that results in damage to the structure of the commensal microbiota and release of oxygen into the colonic lumen (Fig 7). This ultimately allows *Salmonella* to outcompete resident gut microbes by using the host immune response, rather than direct competition for resources in the gut.

Many studies in the fields of immunology and pathogenesis rely on mouse mortality as a measurable readout for virulence. In the current study, we found that *IL-1β* $^{-/-}$ mice do not succumb to *Salmonella* infection. To help us identify the mechanism leading to this resistance, we conducted a literature search to understand why this infection is lethal in C57BL/6 WT mice. Yet, we were surprised that our literature search on the mechanisms that drive mortality during non-typhoidal Salmonellosis did not result in definitive explanations. Death from non-typhoidal Salmonellosis in mice carrying the susceptibility allele in *Nramp1* is attributed to an inability to control intracellular bacteria by macrophages [5]. Yet, we could not find definitive evidence to support this notion. Another study has found that mortality from intravenous infection is dependent on bacterial lipid A and is characterized by robust IL-1β secretion [31]. Yet, the downstream target of IL-1β that leads to mortality was not known. Our discovery that IL-1β dramatically up-regulates the expression of complement proteins and down-regulates the expression complement-inactivator in vivo provides a possible mechanism for *Salmonella*-driven mortality (Fig 7). This mode of regulation can thus maximize the antimicrobial activity of the complement system as it drives complement expression and inhibits its degradation simultaneously.

In humans, a global systematic review and meta-analysis found that the most common reason for non-typhoidal *Salmonella* disease-associated complications and fatalities was septicaemia [32]. The definition of septicaemia is "an infection that occurs when bacteria enter the bloodstream and spread," according to the Cleveland Clinic [33]. As in mice, this explanation does not provide a testable mechanism. Studies in nonhuman primates have shown that inhibition of complement proteins can prevent mortality from intravenous bacterial infection [34]. Complement proteins are known to elevate vascular permeability, cause smooth muscle contraction and drive cardiomyopathy, all of which lead to fatal outcome [35]. Thus, control of complement expression needs to be linked to acute immune activation, while also being tightly regulated to prevent excessive damage. Our finding that IL-1β drives expression of complement anaphylatoxins, and suppresses expression of complement-inactivating CPN1, provides an example where immune regulation of the complement system can spiral out of control, leading to death (Fig 7). This also provides a possible mechanism for non-typhoidal

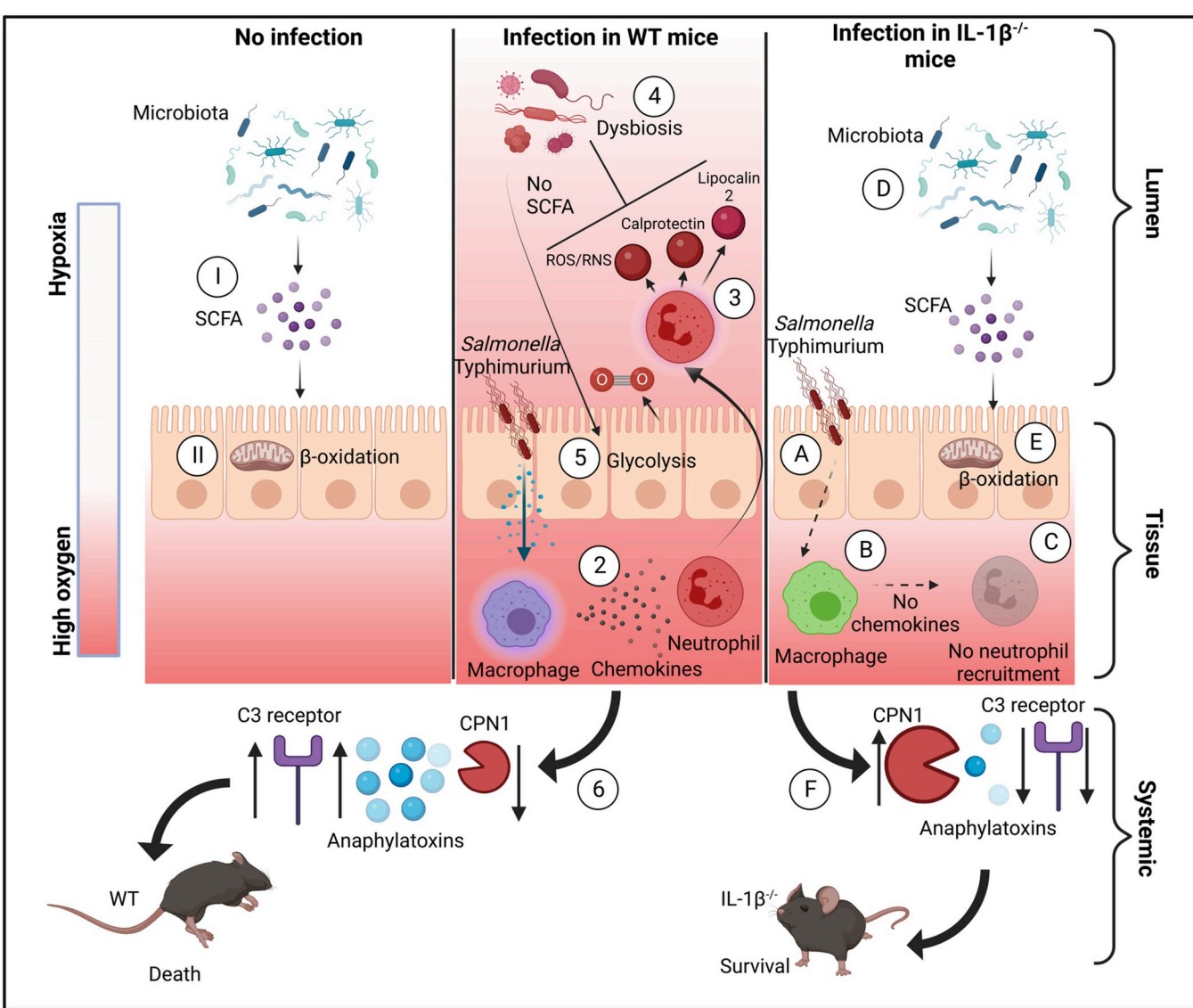

**Fig 7. IL-1β is a key axis in host manipulation by *Salmonella*.** (**I**) In naïve mice, commensal microbes produce SCFA. (**II**) These SCFA are used to produce energy via β-oxidation by colonocytes. As this process uses oxygen, the gut lumen remains hypoxic. (**1**) *Salmonella* infection in WT mice leads to IL-1β secretion and (**2**) production of chemokines that attract neutrophil to the gut lumen. (**3**) These neutrophil produce ROS and NOS and secrete metal chelators which (**4**) kills commensal microbes, thus reducing SCFA levels. This forces colonocytes to produce energy via fermentation which does not uses oxygen (**5**), thus allowing oxygen to seep into the gut lumen, further damaging the microbiota. (**6**) Release of IL-1β following infection leads to elevated levels of anaphylatoxins and suppressed levels of CPN1, leading to mortality. (**A**) In mice lacking IL-1β, *Salmonella* infection (**B**) does not lead to production of chemokines and (**C**) does not attract neutrophils to the gut lumen. (**D**) These preserve SCFA-producing commensals and (**E**) allow colonocytes to produce energy via β-oxidation, which preserves luminal hypoxia which impedes *Salmonella* expansion. (**F**) Lack of IL-1β reduces anaphylatoxin levels and maintains high levels of CPN1 that prevents mortality. CPN1, carboxypeptidase N; SCFA, short-chain fatty acid.

*Salmonella*-induced mortality. While we show a correlation between expression of *IL-1β* and the complement receptor *C3AR1* in humans suffering from sepsis, it will need to be tested whether this is the mechanism which results in fatalities from septicaemia in humans.

Finally, our study presents results in *Salmonella*-infected *IL-1β* [-/-] mice which differ from a previous report by Raupach and colleagues which also used *Salmonella*-infected *IL-1β* [-/-] mice [36]. The work by Raupach and colleagues did not examine luminal bacterial loads in the intestine but found higher bacterial loads in inner tissues of *IL-1β* [-/-] mice as compared to WT

mice. Also, Raupach and colleagues did not find that *IL-1β* [-/-] mice exhibit reduced *Salmonella*-induced mortality. One reason for this discrepancy might be that here we used the streptomycin-depletion model [13], which reduces the influence of the preinfection microbiota (which can vary between animal facilities), while Raupach and colleagues did not pretreat the mice with antibiotics. Indeed, variations in the preinfection microbiota can widely influence infection outcome and mouse survival, especially when antibiotic pretreatment is not performed [14]. Another possible reason for this discrepancy is that Raupach and colleagues used a 10-fold higher inoculation dose compared to the dose we used here. At such high numbers ($10^8$ C.F.U.), the host might be overwhelmed to a point where genetic differences are less influential. Indeed, we also show here that loss of IL-1β does not provide protection from mortality during systemic infection. Finally, the genetic background of the mice might also explain this discrepancy, as is the case in *Caspase 1*-deficient mice [36–38].

## Materials and methods

### Ethics statement

All experiments and procedures involving animals were approved by the institutional animal care and use committee (IACUC) of the Bar-Ilan University (study ID#14-02-2020). This study was performed according to national laws and regulations, and with compliance to EU directive regarding the protection of animals used for experimental and other scientific purposes. The Bar-Ilan IACUC has approved the animal experiments only after ethical balancing of the benefits of the project against the suffering caused to the animals. The approval is firmly based on the principle of the 3 Rs, to replace, reduce, and refine the use of animals used for scientific purposes.

### Mice

C57BL/6 WT mice and *IL-1β* [-/-] mice (backcrossed to C57BL/6 for over 10 generations [12]) were separately bred and maintained in the conventional barrier at the Azrieli Faculty of Medicine, Bar-Ilan University, Israel; 8- to 14-week-old mice were used for all experiments. For co-housing experiments, mice were co-housed for 2 weeks prior to infection, which is a timeframe shown to be sufficient to eliminate microbiota differences [15].

### *Nramp* genotyping

DNA from mouse tails were extracted using a standard genotyping method. The *Nramp* gene was amplified using the following primers: forward- 5′–AAGTGACATCTCGCCATAGGTGCC–3′ and reverse- 5′–TTCTCTCACCATAGTTATCCAAGAAG–3′. The purified PCR product was sequenced using the primer 5′–CCCCCATCTATGTTATCACCC–3′.

### *Salmonella* infection

Mice were treated with 20 mg streptomycin via gavage 24 h before oral infection with $10^7$ C.F. U. *Salmonella* enterica serovar typhimurium (SL1334) or intravenous infection with $10^5$ C.F. U. *Salmonella*. For vancomycin treatment model, mice were treated with 300 mg of streptomycin and 150 mg of vancomycin in drinking water in a volume of 300 ml throughout the course of the infection. At the designated time points, mice were euthanized and cecal content, liver, spleen, and mesenteric lymph nodes were removed, weighed, homogenized in PBS, and plated on LB agar plates containing streptomycin following serial dilutions. For survival assays, mice that were unresponsive to touch, cold or could not reach their food were euthanized.

## Gentamycin protection assay

Mice were treated with 1 ml of thioglycolic acid via intraperitoneal injection and euthanized after 5 days, after which 5 ml of PBS was injected to the peritoneal cavity and mixed gently. The PBS containing cells was then drawn out and stored on ice. The solution was centrifuged for 10 min at 400 g, supernatant discarded, and cell pellet was resuspended with 1 ml DMEM with serum and filtered in 40 μm filter. Filtered cells were counted and $10^6$ cells were plated on cell culture plate overnight at 37˚C. Wells were washed 3 times with PBS (37˚C) and incubated with *Salmonella* at a multiplicity of infection (MOI) = 3 for 90 min. Infected cells were washed 3 times with PBS (37˚C) and incubated with 200 mg/ml gentamicin for 90 min and then washed with PBS. Lysis was then performed with 1% of Triton X-100 for 15 min. The lysate was diluted and seeded on LB agar plates containing streptomycin for overnight incubation.

## RNA sequencing and analysis

RNA from frozen colonic tissues was extracted using Qiagen RNeasy Universal kit. Integrity of the isolated RNA was analyzed using the Agilent TS HS RNA Kit and TapeStation 4200 at the Genome Technology Center at the Azrieli Faculty of Medicine, Bar-Ilan University, and 1,000 ng of total RNA was treated with the NEBNext poly (A) mRNA Magnetic Isolation Module (NEB, #E7490L). RNA sequencing libraries were produced by using the NEBNext Ultra II RNA Library Prep Kit for Illumina (NEB #E7770L). Quantification of the library was performed using a dsDNA HS Assay Kit and Qubit (Molecular Probes, Life Technologies) and qualification was done using the Agilent TS D1000 kit and TapeStation 4200, and 250 nM of each library was pooled together and diluted to 4 nM according to the NextSeq manufacturer's instructions; 1.6 pM was loaded onto the Flow Cell with 1% PhiX library control. Libraries were sequenced with the Illumina NextSeq 550 platform with single-end reads of 75 cycles according to the manufacturer's instructions. Sequencing data was aligned and normalized (reads per million mapped reads) using Partek bioinformatics software. Pathway analysis was performed using the ShinyGO webtool [39]. Heat maps, principal coordinate analysis (PCoA) plots, and volcano plots were generated using GraphPad Prism software. Raw data is deposited in GSE252071.

## Histology, immunohistochemistry, and inflammation scoring

Distal colon tissues were fixed in 4% paraformaldehyde, paraffin embedded, sectioned, and stained with hematoxylin and eosin. Histopathological analysis and semi-quantitative scoring were performed by a board-certified toxicological pathologist according to the scoring system described by Cooper and colleagues [40], taking into consideration the grades of extension (laterally, along the mucosa and deep into the mucosa, submucosa, and/or muscular layers) of the inflammation and ulceration, as previously described [41]. Analysis was performed in a blinded fashion. Immunohistochemistry for neutrophil elastase and myeloperoxidase were performed using Bioss bs-6982R and Abcam ab9535 antibodies, respectively, according to manufacturer's instructions. Positively stained cells were counted in 6 randomly selected fields.

## Blood cytometry and chemistry

Whole blood was drawn via cardiac puncture. Blood samples were collected into a blood collection tube, 50 μl for complete blood count (CBC) and 180 μl for biochemistry. The samples sent at a temperature of 4 degrees to a service and analysis laboratory: American Medical Laboratories (AML) in Herzliya, Israel.

## Microbiota profiling and analysis

Bacterial DNA was extracted from feces, using the Mobio PowerSoil DNA extraction kit (MoBio) following a 2-min bead-beating step (Biospec). The V4 of the 16S rRNA gene was amplified using PCR with barcoded primers. DNA was then purified using AMPURE XP magnetic beads (Beckman Coulter) and quantified using Quant-iT PicoGreen dsDNA Assay (Thermo Fisher) and equal amounts of DNA were then pooled and sequenced. After sequencing on an Illumina MiSeq platform at the Faculty of Medicine Genomic Center (Bar Ilan University, Safed, Israel), single-end sequences reads were import and demultiplexed using QIIME 2 (version 2023.2) [42]. Sequencing errors were corrected by DADA2 [43] and taxonomic classification was done using Greengenes reference database with confidence threshold of 99% [44]. Principal coordinate analysis (PCoA) was performed using Jaccard distances, which calculate the difference between the presence or absence of features [45]. ANCOM was used to identify differentially abundant taxa [46]. Raw data is deposited in GSE252071.

## qRT-PCR

For quantification of Clostridia, bacterial DNA was extracted from feces using the Mobio PowerSoil DNA extraction kit (MoBio) following a 2-min bead-beating step (Biospec). The DNA quantified using Quant-iT PicoGreen dsDNA Assay (Thermo Fisher). The concentrations of the samples were diluted to 100 ng. qPCR was performed with Cyber Green with the following primers:

Pan Bacterial 16s:
Forward: 5′–CCTACGGGAGGCAGCAG–3′,
Reverse: 5′–ATTACCGCGGCTGCTGG–3′.
Clostridia:
Forward: 5′– ACTCCTACGGGAGGCAGC–3–3′,
Reverse: 5′– GCTTCTTTAGTCAGGTACCGTCAT –3′.

For mouse gene expression, RNA was extracted using Qiagen RNeasy Universal kit and cDNA prepared using Thermo High-Capacity cDNA Reverse Transcription Kit. qPCR was performed with Cyber Green with the primers detailed in S1 Table.

## Detection of hypoxia in vivo

Mice were treated intraperitoneally with a 1.2 mg solution of pimonidazole HCl (Hypoxyprobe Kit) 1 h before euthanasia. Distal colon tissues were fixed in 4% paraformaldehyde, paraffin embedded, sectioned, and stained with of 11.23.22.r Rat Mab for 1 h. Sections were visualized with a Zeiss Axio imager M2 according to manufacturer's instructions. Scoring was performed in a blinded fashion.

## Carboxypeptidase-inhibitor treatment

Mice were infected orally with *Salmonella* as above. After infection, mice were treated with 1.25 mg of DL-Benzylsuccinic acid (in 10% ethanol) via intraperitoneal injection every 8 h. Control mice were treated with 10% ethanol in PBS. Mice were physically examined before and after injections to exclude damage due to the procedure. Mice hurt during the injection were excluded from the experiment.

## Statistical analysis

Results were analyzed using Graph-Pad Prism 10 software (GraphPad, La Jolla, California, United States of America).

## Supporting information

**S1 Fig. *IL-1β*** [superscript -/-] **mice are *Nramp*** [superscript S]**.** Sequencing of genomic DNA from CBA mice (*Nramp*[r]), WT C57BL/6 mice (*Nramp*[S]), and and *IL-1β*[-/-] mice on a C57BL/6 background (*Nramp*[S]). The red rectangle highlights the sensitivity mutation G->A.
(TIFF)

**S2 Fig. *IL-1β*** [superscript -/-] **mice co-housed with WT mice are resistant to *Salmonella* infection.** (**A–D**) *Salmonella* C.F.U. 4 days post-infection in cecal content (**A**), M.L.N. (**B**), liver (**C**), and spleen (**D**) of co-housed mice infected with $10^7$ C.F.U. *Salmonella* enterica serovar typhimurium (SL1334) 24 h after pretreatment with 20 mg streptomycin. (**E**) Survival percentage of mice infected orally with *Salmonella*. (**A–D**) Each dot represents a mouse. These data are representative of 1 experiment. $^*P < 0.05$; $^{**}P < 0.01$; $^{****}P < 0.0001$. (**A–D**) Mann–Whitney test. (**E**) Mantel–Cox test. C.F.U., colony-forming units; Sep, separately housed; Co, co-housed; d.p.i., days post-infection. Numerical values are in S1 Data.
(TIFF)

**S3 Fig. Macrophages from *IL-1β*** [superscript -/-] **mice are compromised in their ability to clear intracellular *Salmonella*.** (**A**) Gentamycin protection assay using peritoneal macrophages extracted from mice infected with *Salmonella* ex vivo with a MOI of 3. These data are representative of 1 experiment. $^*P < 0.05$; Student's *t* test. Numerical values are in S1 Data.
(TIFF)

**S4 Fig. Loss of IL-1β does not affect transcript levels of *Lcn2* and *Cxcl9* in naïve mice.** (**A** and **B**) qPCR analysis of (**A**) *Lcn2* and (**B**) *Cxcl9* transcripts in colons of naïve mice. Expression was normalized to 18S. These data are representative of 1 experiment. ns, not statistically significant; RQ, relative quantity. Student's *t* test. Numerical values are in S1 Data.
(TIFF)

**S5 Fig. Clostridia are not depleted in infected *IL-1β*** [superscript -/-] **mice.** (**A**) Absolute number of Clostridia OTUs in gut microbiota of *Salmonella*-infected mice as in Fig 3. (**B**) qPCR analysis of levels of the class Clostridia in uninfected mice housed and treated as indicated. Each dot represents a mouse. These data are representative of 1 experiment. $^*P < 0.05$. Student's *t* test. OTUs, operational taxonomic unit; RQ, relative quantity. Numerical values are in S1 Data. The underlying data for this figure can be found at GSE252071.
(TIFF)

**S6 Fig. Loss of IL-1β does not affect transcript levels of metabolism genes in naïve mice.** qPCR analysis of (**A and B**) transcripts involved in fatty acid metabolism and (**C–E**) transcripts involved in glycolysis in colons of naïve mice. Expression was normalized to 18S. These data are representative of 1 experiment. ns, not statistically significant; RQ, relative quantity. Student's *t* test. Numerical values are in S1 Data.
(TIFF)

**S7 Fig. Vancomycin treatment depletes SCFA-producing *Clostridia* in mice.** 16S rRNA sequencing was performed to characterize gut microbiota composition of mice treated with vancomycin for 3 days. Relative abundance (left) and absolute OTU reads (right) of SCFA-producing members of the Clostridia class at the family level. Each symbol represents a mouse. OTU, operational taxonomic unit. These data are representative of 1 experiment. $^{**}P < 0.01$; Student's *t* test. Numerical values are in S1 Data.
(TIFF)

**S8 Fig. Vancomycin-mediated depletion of SCFA-producing bacteria does not affect survival of *IL-1β* $^{-/-}$ mice.** (**A**) Survival of vancomycin-treated mice infected orally. (**B**) *Salmonella* C.F.U. in the indicated organs of *IL-1β* $^{-/-}$ mice 17 d.p.i. Each dot represents a mouse. (**C**) Survival of WT mice infected with *Salmonella* isolated from *IL-1β* $^{-/-}$ mice 21 d.p.i. \*\*$P < 0.01$. (**A**) Mantel–Cox test. These data are representative of 1 experiment. d.p.i., days post-infection. Numerical values are in S1 Data.
(TIFF)

**S9 Fig. Treatment with carboxypeptidase inhibitor is not lethal in mice and does not affect *Salmonella* growth.** (**A**) Survival of uninfected mice treated as in Fig 5. (**B**) Growth curve of *Salmonella* treated as indicated. These data are representative of 1 experiment. Numerical values are in S1 Data.
(TIFF)

**S1 Table. Primers used for qPCR.**
(XLSX)

**S1 Data. Numerical values used to generate graphs and charts in figures.**
(XLSX)

## Acknowledgments

This study was performed in memory of Ron N Apte. We would like to thank Dr. Marina Bersudsky for her assistance in conducting the experiments.

## Author Contributions

**Conceptualization:** Mor Zigdon, Michal Werbner, Omry Koren, Sebastian E. Winter, Ron N. Apte, Elena Voronov, Shai Bel.

**Data curation:** Mor Zigdon, Lilach Zelik, Dana Binyamin, Michal Werbner, Sebastian E. Winter, Ron N. Apte, Elena Voronov, Shai Bel.

**Formal analysis:** Mor Zigdon, Michal Werbner, Elena Voronov, Shai Bel.

**Funding acquisition:** Sebastian E. Winter, Shai Bel.

**Investigation:** Mor Zigdon, Jasmin Sawaed, Shira Ben-Simon, Nofar Asulin, Rachel Levin, Sonia Modilevsky, Maria Naama, Shahar Telpaz, Elad Rubin, Aya Awad, Wisal Sawaed, Sarina Harshuk-Shabso, Meital Nuriel-Ohayon, Mathumathi Krishnamohan, Michal Werbner, Elena Voronov, Shai Bel.

**Methodology:** Mor Zigdon, Michal Werbner, Sebastian E. Winter, Ron N. Apte, Elena Voronov, Shai Bel.

**Project administration:** Michal Werbner, Elena Voronov, Shai Bel.

**Resources:** Sebastian E. Winter, Ron N. Apte, Elena Voronov.

**Supervision:** Michal Werbner, Ron N. Apte, Elena Voronov, Shai Bel.

**Writing – original draft:** Mor Zigdon, Michal Werbner, Omry Koren, Sebastian E. Winter, Elena Voronov, Shai Bel.

**Writing – review & editing:** Mor Zigdon, Michal Werbner, Omry Koren, Sebastian E. Winter, Elena Voronov, Shai Bel.

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
