## [Editor Report · Decision Letter 0]

19 Dec 2023

Dear Shai, 

Thank you for submitting your manuscript entitled "Salmonella manipulates the host to drive pathogenicity via induction of interleukin 1β" for consideration as a Research Article by PLOS Biology. I've now had time to assess your revised manuscript and the responses to the Immunity reviewers, and discuss the whole file with one of our academic editors and I am writing to let you know that we would like to move to acceptance of your work after minor (textual) revisions and checks for compliance with our journal policies.

However, before I can send you a decision letter detailing what we'd require for publication, we need you to complete your submission by providing the metadata that is required for full assessment. To this end, please login to Editorial Manager where you will find the paper in the 'Submissions Needing Revisions' folder on your homepage. Please click 'Revise Submission' from the Action Links and complete all additional questions in the submission questionnaire. 

Once your full submission is complete, your paper will undergo a series of checks in preparation for peer review. After your manuscript has passed the checks it will be come to my desk and I will be able to proceed. To provide the metadata for your submission, please Login to Editorial Manager (https://www.editorialmanager.com/pbiology) within two working days, i.e. by Dec 21 2023 11:59PM.

This seems a bit nonsensical in this case, but stems for our easy drag-n-drop initial submission process, because we'd need this information and eg high-quality figures for peer review, and all submissions have to go through this - it should take no more than a few minutes to complete. If your manuscript lands back on my desk by EOB Thursday, then I will be bale to issue a formal decision on Friday morning - otherwise this will happen when I am back in the office in early January.

Feel free to email me if you have any queries relating to your submission, or to plosbiology@plos.org if you have problems with the "full submission" process.

Kind regards,

Nonia

Nonia Pariente, PhD, 

Editor-in-Chief

PLOS Biology

npariente@plos.org

---

## [Editor Report · Decision Letter 1]

22 Dec 2023

Dear Shai,

Thank you for your patience while we considered your revised manuscript "Salmonella manipulates the host to drive pathogenicity via induction of interleukin 1β" for publication as a Research Article at PLOS Biology. This is a revised manuscript addressing the issues raised by reviewers at another journal, and you have provided us with the revision and the point by point rebuttal to the reviewer concerns. I have now had time to assess the whole file, as well as to discuss it with an Academic Editor of relevant expertise. Based on this assessment, we are likely to accept this manuscript for publication, provided you satisfactorily address the points indicated below:

- Please expand the rather succinct introduction and discussion section to include the relevant aspects of your responses to reviewers, including a discussion of the research context and how your study adds to the existing literature (ie why it is novel, which we agree it is. You need to spell this out). Additionally, you should directly discuss Zychlinsky's paper (no increased mortality in IL1B KO mice) and why you came to a different conclusion. We have no problem with that, as you are using a completely different model (strep pretreatment).

- Please add to each supplementary figure legend (except Sup Fig 1) information about how many independent experiments were performed

- The Ethics statement needs to be a separate, independent (and the first) subheading in the Material & Methods section. It needs to include not only the full name of the IACUC/ethics committee that reviewed and approved the animal care and use (which you have) but also the protocol/permit/project license number (which is missing).

Please include the specific national or international regulations/guidelines to which your animal care and use protocol adhered.

- DATA POLICY: You may be aware of the PLOS Data Policy, which requires that all data be made available without restriction: http://journals.plos.org/plosbiology/s/data-availability. For more information, please also see this editorial: http://dx.doi.org/10.1371/journal.pbio.1001797

Regardless of the method selected, please ensure that you provide the individual numerical values that underlie the summary data displayed in the main and supplementary figure panels that show data (that is, all of them except Fig 7 and Supp Fig 1), as they are essential for readers to assess your analysis and to reproduce it:

NOTE: the numerical data provided should include all replicates AND the way in which the plotted mean and errors were derived (it should not present only the mean/average values). It should be clearly labelled.

**Please also ensure that all main and supplementary figure legends in your manuscript include information on where the underlying data can be found, eg "The underlying that for this figure can be found at [either a repository URL or the specific Supp File where that specific data is]". Please ensure your supplemental data files have a legend.**

- Data Accessibility Statement (DAS): You will need to deposit all RNAseq data and 16s data to GEO at this time, as the accession number needs to be included in the manuscript before final acceptance (the information can be embargoed - but then you need to provide us with a reviewer token for access, so we can access the information before acceptance).

In the new DAS, please indicate not only the accession numbers for the RNAseq and 16s data, but also where the data underlying the main and supplementary figures can be found, be it a repository or specific supplementary data files.

We expect to receive your revised manuscript by 10 January. 

*Published Peer Review History*

Please note that you will have the opportunity to make the peer review history publicly available. The record will include editor decision letters (with reviews) and your responses to reviewer comments. We will contact you to opt in or out. Please see here for more details:

*Press*

Sincerely,

Nonia

Nonia Pariente, PhD, 

Editor-in-Chief,

npariente@plos.org,

PLOS Biology

---

## [Editor Report · Decision Letter 2]

5 Jan 2024

Dear Shai,

Happy new year - or at least one that improves quickly.

Thank you for the submission of your revised Research Article "Salmonella manipulates the host to drive pathogenicity via induction of interleukin 1β" for publication in PLOS Biology. On behalf of my colleagues and the Academic Editor, Ken Cadwell, I am pleased to say that we can in principle accept your manuscript for publication, provided you address any remaining formatting and reporting issues. These will be detailed in an email you should receive within 2-3 business days from our colleagues in the journal operations team; no action is required from you until then. Please note that we will not be able to formally accept your manuscript and schedule it for publication until you have completed any requested changes.

I have noted that the GEO accession is embargoed until Dec 1, 2024 - please modify this once you become aware f the publication date. As I see that you have opted into early article posting, we will not be able to schedule publication; it will be live 2ish days after you return the proofs

Please take a minute to log into Editorial Manager at http://www.editorialmanager.com/pbiology/, click the "Update My Information" link at the top of the page, and update your user information if needed to ensure an efficient production process.

PRESS

Best,

Nonia

Nonia Pariente, PhD, 

Editor-in-Chief

PLOS Biology

npariente@plos.org